# Solid-state $^{31}$P NMR reveals the biological organophosphorus compounds as the dominant phosphorus species in Saharan dust aerosols

Kalliopi Violaki [1] ✉, Christos Panagiotopoulos [1,2] ✉, Claudia Esther Avalos [3,4], Pierre Rossi[5], Ernest Abboud[1], Maria Kanakidou [6,7,8] & Athanasios Nenes [1,7]

Phosphorus is a critical nutrient affecting primary productivity across all ecosystems. Many regions worldwide are limited or co-limited by phosphorus availability, which can be alleviated through atmospheric deposition. Dust is known to be a key external source of phosphorus in ecosystems, assumed to be in the form of various insoluble inorganic minerals. We show that this view is largely incomplete and here we present conclusive evidence, that organic phosphorus as diesters, primarily associated with biological materials. Phosphate diesters significantly correlated with soil bacteria found in dust, implying a direct link with microbial soil communities, without excluding the eukaryotic cells. Phosphate diesters in dust, along with abundant alkaline phosphatase, may contribute 70% to daily primary productivity in the eastern Mediterranean, highlighting the potential of organic phosphorus substrates present in dust as airborne microorganisms to impact the biogeochemistry of oligotrophic environments via atmospheric deposition.

Phosphorus (P) is a key element for all living organisms; making it the most important nutrient across aquatic and terrestrial ecosystems[1–3]. Being the basic component of DNA and RNA, it serves as a structural component of cell membranes in several organisms (e.g., phospholipids)[4,5] and is central to several biological processes (e.g., energy transactions via ATP). It exists in the environment in inorganic (i.e., orthophosphate or $PO_4^{3-}$) and organic forms, of which the latter embraces not only a wide variety of biological molecules, such as phospholipids, phosphonates, sugar phosphates, nucleotides, nucleic acids, and some vitamins[2] but also organophosphorus compounds with anthropogenic origin, such as flame retardants and pesticides[6–9].

In marine environments, dissolved organic P (DOP) dominates the total dissolved P (TDP) pool, representing 50%–60% in surface waters (0–100 m) with DOP values within the range of 0.1–0.4 μM[2,10]. In ultraoligotrophic systems (e.g., the Mediterranean Sea), surface DOP

values are generally one order of magnitude lower (0.04–0.06 μM)[11] than other oceanic regimes, whereas surface inorganic P values are in the low nM level[12].

The current scientific view of the atmospheric P associated with airborne particles[13] is that inorganic species (orthophosphate, pyrophosphate, and polyphosphate), which are mainly present in mineral dust, are present in substantially higher amounts than organic P species[14,15]. Most atmospheric P studies focus on the inorganic component and the role of atmospheric processing on its bioavailability[16–18] and potential to fertilize ultraoligotrophic areas, whereas the organic counterpart has received much less attention than it deserves[19–22]. Previous investigations that assessed organic P composition in atmospheric particles in the Mediterranean using near-edge X-ray fluorescence spectroscopy revealed that the spectral linear combination fitting of pollen and bacteria was dominated by the organic phosphorus fraction; however, further research is warranted to characterize

[1]Laboratory of Atmospheric Processes and their Impacts, School of Architecture, Civil & Environmental Engineering, École Polytechnique Fédérale de Lausanne, Lausanne, Switzerland. [2]Aix Marseille Univ., Université de Toulon, CNRS, IRD, MIO, Marseille, France. [3]Institute of Chemical Sciences and Engineering, NMR platform, EPFL, Rte Cantonale, Lausanne, Switzerland. [4]Department of Chemistry, New York University, New York, NY, USA. [5]Central Environmental Laboratory, School of Architecture, Civil & Environmental Engineering, École Polytechnique Fédérale de Lausanne, Lausanne, Switzerland. [6]Environmental Chemical Processes Laboratory (ECPL), Department of Chemistry, University of Crete, Heraklion, Greece. [7]Center for the Study of Air Quality and Climate Change (C-STACC), Institute of Chemical Engineering Sciences, Foundation for Research and Technology, Patras, Greece. [8]Institute of Environmental Physics, University of Bremen, Bremen, Germany. ✉e-mail: kalliopi.violaki@epfl.ch; christos.panagiotopoulos@mio.osupytheas.fr

the organic phosphorus-containing phases[23]. The molecular level analyses based on mass spectrometry techniques showed that compounds containing P were either organic contaminants[6,7,9] or phospholipids originating from bioaerosols[22]. Finally, it is worth mentioning that organic P in marine and atmospheric studies is not directly determined but is always inferred by subtracting all inorganic species from total P (TP). This is achieved by converting all forms of P to orthophosphate (by UV or persulfate oxidation) and subsequent colorimetric analysis. Therefore, errors related to the efficiency of this transformation may also affect the amount of organic P[2]. Considering the importance of organic P in the atmosphere[22], it is critical to directly quantify its pool and fully assess the real contributions among different organic P species.

Solid-state $^{31}$P NMR is a powerful technique that identifies phosphorus chemical species, organic or inorganic, without any previous sample treatment. This technique was widely used in the 2000s for marine studies in samples comprising concentrated dissolved organic matter fast-sinking particles, marine plankton, and sediments, revealing important features of organic P compounds such as phosphonates[24–26]; however, it was not popular in atmospheric P studies. The main goal of this study is to explore the use of solid-state $^{31}$P NMR spectroscopy for the determination and quantification of organic P-containing compounds present in the Saharan dust, understand its origin, and assess whether it can be an important pool of P compared to the established inorganic counterpart. Furthermore, the role of airborne bioaerosols as a potential regulator of primary productivity is investigated.

## Materials and methods
### Sampling
Saharan dust aerosols were collected from the eastern Mediterranean by a high-volume TSP sampler (TISCH) on precombusted (450 °C for 5 h) 20 × 25 cm quartz filters (Pall, 2500QAT-UP). The sampling resolution was 48 h at a flow rate of 85 m³ h⁻¹. More details on the sampling area are given elsewhere[22]. The samples were collected in March 2018, a typical month of dust episodes in the area, with an exceptionally strong dust event on March 22, 2018.

### Solid-state $^{31}$P NMR measurements
$^{31}$P magic-angle spinning (MAS) NMR spectra were recorded on 400 MHz (9.4 T) and 500 MHz (11.7 T) Bruker spectrometers equipped with an Avance III HD console and 3.2 mm three-channel low-temperature MAS probes. Each dust sample was collected by gently scraping the surface of the quartz filter with a clean, sterilized aluminum spoon, avoiding detaching the quartz fibers during the process. Samples were packed into 3.2 mm zirconia rotors, which were weighed before and after packing. The samples were handled in the air at room temperature. The rotors were spun at 24 kHz spinning using dry nitrogen gas at ambient temperature without temperature regulation. $^{31}$P MAS NMR spectra were recorded using simple one-pulse or Hahn echo sequences. Pulse widths ($\tau_{90}$) ranged from 3.0 to 4.5 µs, depending on the probe and its configuration. Echo delays were set to one rotor period. Between 9024 and 55,296 transients were accumulated for every sample (this corresponds to ~2–3 days of NMR run), except for the reference NH$_4$H$_2$PO$_4$, for which 2 scans were sufficient. Recycle delays were chosen to be at least 5 × T$_1$ whenever T$_1$ was known. In the absence of a signal, a maximal T$_1$ of 5 s was assumed. All experimental parameters, except the number of transients, were kept the same for all the samples, including the reference. $^{31}$P chemical shifts were referenced relative to 85% H$_3$PO$_4$ using the secondary reference NH$_4$H$_2$PO$_4$ at 1.33 ppm[27].

Fits were performed using the dmfit program[28] (https://nmr.cemhti.cnrs-orleans.fr/dmfit/default.aspx). NH$_4$H$_2$PO$_4$ was used as an external standard, and integrals of the fitted model were used for the quantification of all P species (orthophosphate, P-diesters, and pyrophosphate), considering a linear dependency of the peak area to the number of $^{31}$P nuclei in the sample (Fig. 1). Sample fits were performed using a Gaussian line-shape model, and average amplitude errors after dmfit simulation were <5% for all P species. Finally, although dust samples comprised several paramagnetic species including iron, their effect on $^{31}$P NMR signal was not considerable aside from some noticeable broadening.

### Chemical and biological analyses
Dust aerosols were analyzed for main anions (Cl⁻, NO$_3$⁻, SO$_4$²⁻, HPO$_4$²⁻, and C$_2$O$_4$²⁻) and cations (Na⁺, NH$_4$⁺, K⁺, Mg²⁺, and Ca²⁺), TP, elemental

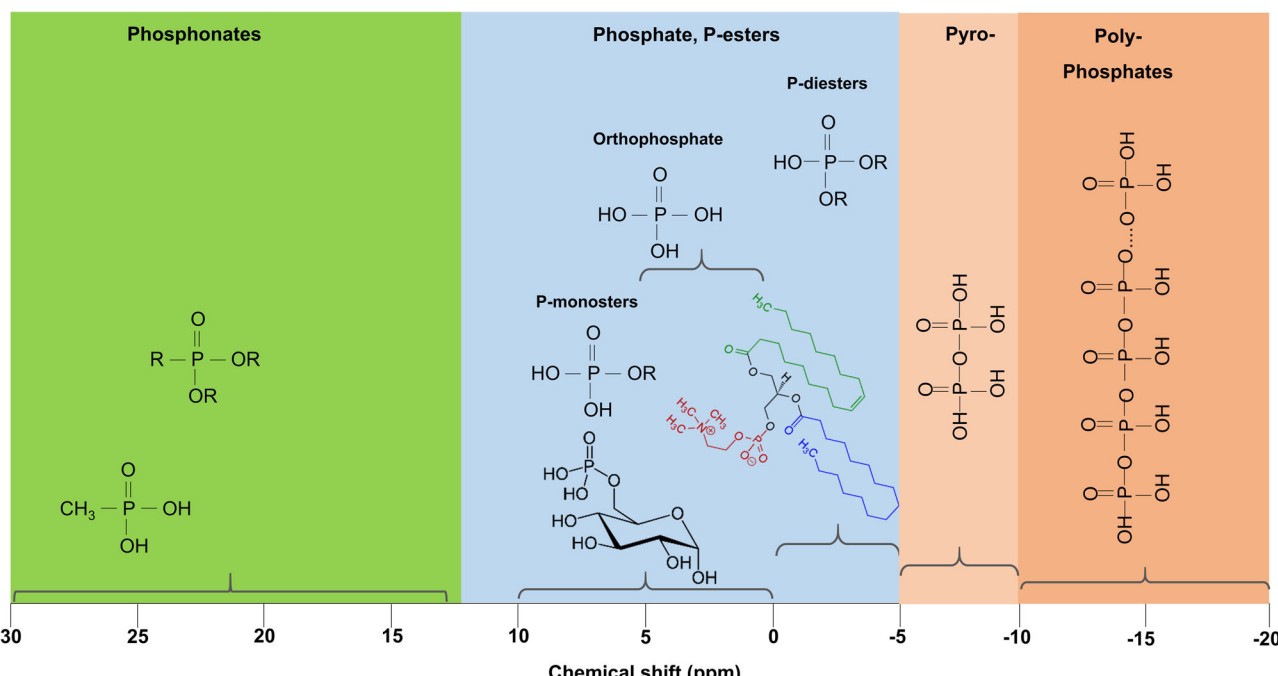

**Fig. 1 | Examples of environmental P compounds and their corresponding chemical shift ranges for $^{31}$P NMR.** Compounds across this range of chemical shift include phosphonates (C-P bonds; e.g methyl phosphonic acid); phosphate esters (R-O-P bonds) which include phosphate monoesters (e.g., glucose 6-P, phytic acid, adenosine monophosphate), orthophosphate (e.g., phosphoric acid), phosphate diesters (phospholipids, Nucleotides, flame retardants), and finally pyrophosphate (P-O-P bonds) and polyphosphate (-O-(PO$_3$)$_n$-).

**Table 1 | P speciation with $^{31}$P NMR and other analytical techniques for the different P-species found in the dust aerosol samples**

| Collection date | Sample ID[a] | $PO_4^{3-}$(%) | Phosphate Diester (%) | Pyrophosphate (%) | TP ($\mu g\ mg^{-1}$) | TP ($\mu g\ mg^{-1}$) |
|---|---|---|---|---|---|---|
| | | $^{31}$P NMR | | | $^{31}$P NMR | ICP-MS |
| 02-04/03/18 | Dust-98 (25.1) | 10 | 88 | 2 | 0.58 | 0.58 |
| 06-08/03/18 | Dust-99 (26.1) | 12 | 87 | 1 | 0.56 | 0.61 |
| 17-19/03/18 | Dust-102 (6.1) | 10 | 82 | 8 | 1.09 | 0.70 |
| 21-23/03/18 | Dust-103 (25.7) | 19 | 78 | 3 | 0.87 | 0.75 |
| 26-27/03/18 | Dust−106 (26.1) | 16 | 81 | 3 | 0.70 | 0.79 |

The amount of total P (TP) estimated by $^{31}$P NMR and ICP-MS is also given. Values for $^{31}$P NMR correspond to percentages of peak areas calculated by dividing their areas by the total spectral peak area of the sample (see Fig. 2). TP is the amount of total P in μg per mg of sample.
[a]In parenthesis is the amount of sample in mg disposed for NMR analysis.

analysis (Al, Ca, Mn, Fe, V, Cr, Ni, Cu, Zn, Cd, and Pb), and phospholipids (PLs); here, they were used for statistical purposes. The total PLs is the sum of main phosphatidylcholines, phosphatidylethanolamines, and phosphatidylglycerols. The analytical protocols described by Violaki et al. [22].

DNA extraction and qPCR analysis were performed in dust aerosol samples for determining the number of bacteria (as copies of the 16S rRNA gene) and fungi (as copies of the 18S rRNA gene) per gram of sample. The details of the protocols are described in Supplementary Section S1 (Supplementary Table S4).

### Reporting summary
Further information on research design is available in the Nature Portfolio Reporting Summary linked to this article.

## Results and Discussion
### P speciation by solid-state $^{31}$P NMR in atmospheric samples and comparison with the aquatic ecosystem
The speciation of P in the Saharan aerosol samples by $^{31}$P NMR (Table 1) revealed the presence of three functional groups, namely, orthophosphate (1.2–1.5 ppm), P-diesters (0.3 − 6 ppm), and pyrophosphate (−8.5 to −13.5 ppm) (Fig. 2, Supplementary Fig. S2), following the same range as those reported in the literature[24]. P-monoesters (0–10 ppm) could not be deconvoluted by $^{31}$P NMR as they overlap with the orthophosphate species (0–5 ppm). P-monoester-related species (e.g., triethyl phosphate) have been reported recently in atmospheric samples as the main components of plants contributed to both gas and particulate phases[29]. However, orthophosphate anions ($PO_4^{3-}$) measured by ion chromatography showed that the average percentage contribution of $PO_4^{3-}$ to TP was similar in both techniques (13%) (Supplementary Table S1), supporting the assumption that the $^{31}$P NMR signal of monoesters is not dominated against the orthophosphate signal in the studied Saharan dust.

The cumulative results of the dust samples showed the predominance of P-diesters, accounting for 83% (78%–88%) of TP, followed by orthophosphates (10%–19% of TP) and pyrophosphates (1%–8% of TP) (Table 1). P-diester bonds are mainly found in phospholipids and nucleotides, highlighting an important biological source of atmospheric organic P. The biological material (bioaerosols) emitted into the atmosphere primarily originates from pollen, microorganisms, cell fragments, and plant debris[30]. These results agree with previous observational and modeling studies indicating that bioaerosols contribute substantially to the atmospheric organic P fraction[22,31].

In the marine environment, the high-molecular-weight dissolved organic matter (HMWDOM) samples (powder obtained after ultrafiltration) assessed by $^{31}$P NMR showed a relatively similar ratio of P-esters (75%; mono and diesters) and phosphonates (25%) regardless of the location or depth of sampling[24]. Phosphonates were identified as the dominant species of organic P in coral tissues[32]; they accounted for 6%–20% of TP in marine sediments[26,33]. However, more recently, a plethora of phosphonic acids including their methyl and hydroxymethyl derivatives were identified in bacterial cultures and HMWDOM samples by high-resolution mass

spectrometry and 2D $^{31}$P–$^1$H heteronuclear multiple bond correlation liquid-state NMR[34]. Phosphonates in seawater may originate from organic matter processed by marine bacteria and archaea and can further be utilized by marine organisms, producing methane[34,35]. Surprisingly, no phosphonates were detected in the dust aerosol samples (Table 1), indicating the absence of C–P bonds in organic molecules containing P. Condensed P forms such as polyphosphates have also been reported in surface marine sediments[26] and sinking particles[25], accounting for <10% of TP, and are generally associated with the storage of P in microorganisms[36]. Polyphosphates were not detected in the studied dust aerosol samples. The solid-state $^{31}$P NMR measurements showed only the presence of pyrophosphates, accounting for <10% of TP (Table 1). The presence of pyrophosphates in the samples may point to the presence of fragments of ATP originating from living cells[37–39].

Finally, TP quantification by $^{31}$P NMR agreed quite well with the ICP-MS measurements (Table 1). The amount of P estimated by $^{31}$P NMR in the dust samples ranged from 0.5 to 1 μg per mg of sample, indicating that $^{31}$P NMR is a potential alternative to quantify P compared to ICP-MS in the condition that a few μg of P is present in the sample. The average TP concentration estimated by ICP-MS was $3.4 \pm 0.6$ nmol m$^{-3}$ ($n = 5$) and agreed with the values recorded during intense dust events in the same area[22]. The average concentrations ($n = 5$) for the individual P species comprising organic P, $PO_4^{3-}$, and condensed P were $2.2 \pm 0.5$, $0.9 \pm 0.7$, and $0.3 \pm 0.3$ nmol m$^{-3}$, respectively.

### Sources of atmospheric P-diesters ($R_1$–O–P–O–$R_2$)
The above results clearly show that organic P compounds in the Saharan dust aerosol are dominated by P-diesters ($R_1$–O–P–O–$R_2$). This functional group is abundant in phospholipids, deoxyribonucleic (DNA), ribonucleic acids (RNA), and their derivatives, linking these compounds to an important biological source of atmospheric organic P[40]. The statistical analysis based on the Pearson correlation ($r_p$) (Supplementary Table S2) revealed that the P-diester fraction is correlated significantly not only with crustal elements such as Mn, K, $Ca^{2+}$, and Fe but also with Pb, As, Ni, Cr, and $SO_4^{2-}$, which are tracers of anthropogenic emissions. These correlations imply the contribution of soil dust to P-diester sources, which is enriched with anthropogenic components during its atmospheric residence and interaction with pollution over the populated areas of North Africa, without excluding the urban imprints on airborne bacterial communities[41].

P-diesters significantly correlated with bacteria (as copies of the 16S rDNA gene per gram of sample, ($r_p = 0.9$, $p < 0.05$, and $n = 5$; Supplementary Table S2), implying a direct link with the microbial communities detected mainly in dust soil (Fig. 3). The majority (90%, $n = 5$) of bacteria communities recorded in all samples were *Actinobacteria* (31% ± 9%) followed by *Proteobacteria* (25% ± 3%), *Firmicutes* (12% ± 2%), *Cyanobacteria* (10% ± 8%), *Bacteroidetes* (7% ± 1%), and *Chloroflexi* (6% ± 3%). These results are consistent with previous studies in Mediterranean aerosols collected during dust events[42–45] or Sahara desert soil[46].

The presence of *Actinobacteria*, *Proteobacteria*, and *Firmicutes* in the dust samples reflects an important contribution of the soil source containing

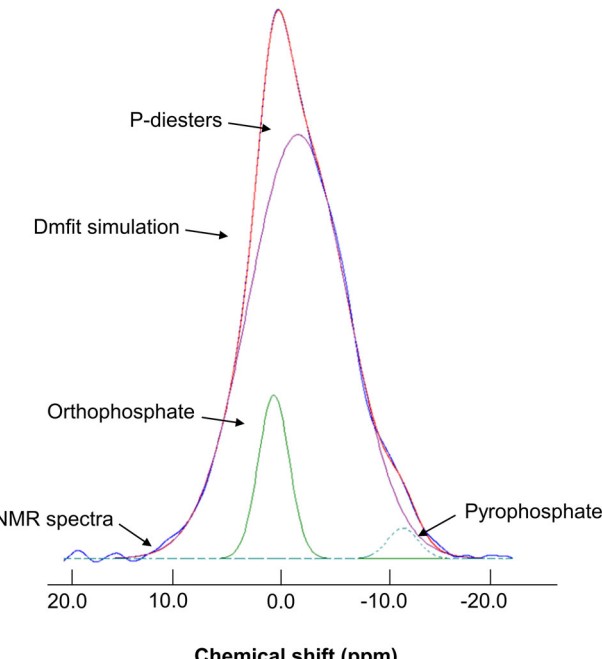

**Fig. 2 | Typical spectra of 31P NMR of a dust sample.** 31P NMR spectra (blue line) of a dust sample (Dust- 98) and the dmfit Monte Carlo simulation (red line). Note that the blue line of 31P NMR spectrum of the sample is overlaid by red line of the Monte Carlo simulation. Signal deconvolution revealed the presence of three functional P groups namely: orthophosphate (green line). P-diesters (violet line) and pyrophosphate (dash line). Chemical shifts of the above-mentioned functional groups fall into the range presented in Fig. 1.

these microorganisms[47], with a minor contribution of the aquatic environment based on the presence of *Cyanobacteria*[47,48]. These results were further confirmed by the Hysplit back trajectory analysis, which showed that air masses mainly originated from the northwestern part of the Sahara desert (Fig. S1, Supplementary Table S3). The significant correlation of P-diesters with PLs ($r_P = 0.9$, p < 0.05, and $n = 4$; Supplementary Table S2) further confirmed the presence of biological material other than bacteria that could contribute to the P-diesters such as eukaryotic cells and plants[22]. Furthermore, Fig. 4 displayed the percentage of identified fungi Orders. The majority of those fungi belong to the *Leotiomycetes* (32 ± 26%), an Order which includes several plant and animal pathogens[49]. In addition, other Order grouping plant pathogens were identified such as *Agaricomycetes* (3 ± 5%)[50] and *Ustilaginomycetes* (3 ± 3%)[51]. Additionally, the majority of identified plant Orders (Supplementary Fig. S3), belongs to the *Brassicales* (25 ± 20%) and the *Fabales* Order (18 ± 28%), which include also species from agriculture activities[52].

## Biogeochemical implications in marine ecosystems

African dust emissions are a significant external P nutrient source not only for the terrestrial but also for the marine environments[53]. Currently, global models predict dust deposition estimates of 1000 to 2000 Tg year$^{-1}$, varying substantially from year to year[54]. Desert sources, as external nutrient paths, are particularly important to the oceans of the northern hemisphere than to the southern hemisphere[53]. However, a recent study supports that one-third of Southern Ocean productivity is supported by dust deposition[55].

It is well known that orthophosphate is the preferred substrate for microorganisms[2]; however, some specific microorganisms favor the utilization of ester-linked P compounds[56,57]. The bioavailability of most organic P pool depends on ambient orthophosphate concentrations, activating the expression of specific enzymes that control transport, and substrate hydrolysis[2]. A well-known example is the alkaline phosphatase, a ubiquitous enzyme in prokaryotic and eukaryotic marine microbes, which hydrolyzes

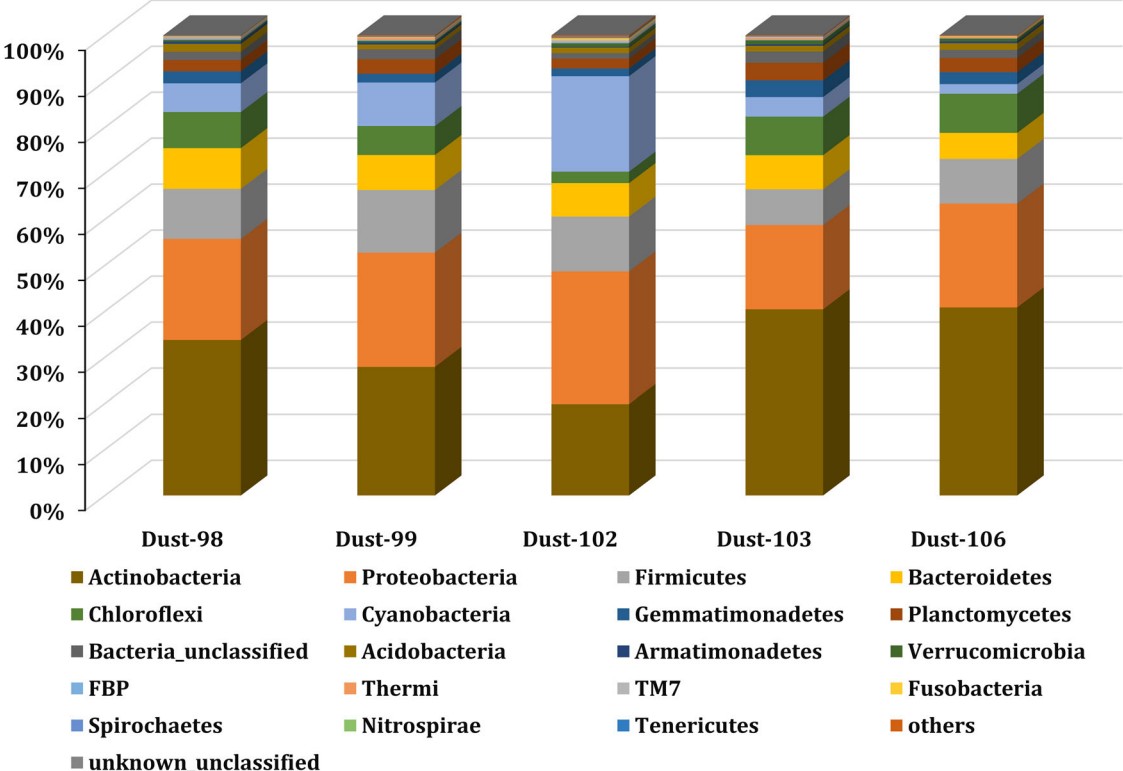

**Fig. 3 |** Relative abundance of bacterial phyla in the dust samples.

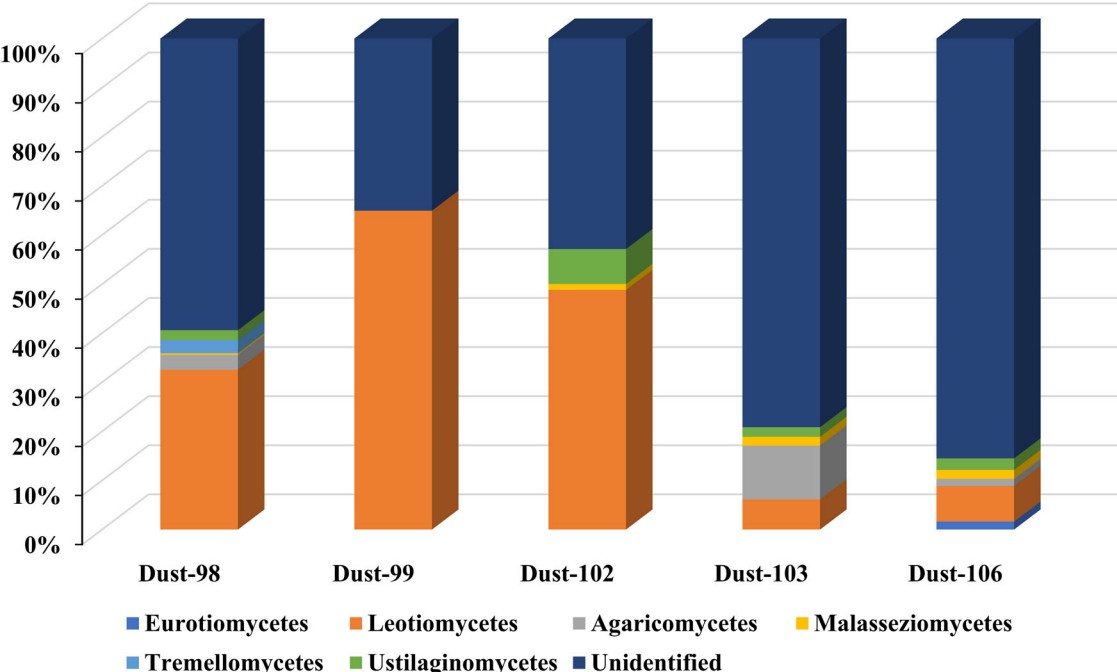

**Fig. 4 |** Relative abundance of fungi order in the dust samples.

P-diesters/monoesters to orthophosphate[58]. Phosphatase activity increases as a result of orthophosphate deprivation, notably in the eastern Mediterranean[59–61], while in the western Mediterranean, both inorganic P and DOP concentrations in seawater control alkaline phosphatase activity[62]. It is worth noting that recent experimental studies showed that some marine cyanobacteria can directly grow on organic phosphorus as the sole P source, which further underlines the importance of the supply of organic P from external sources in the marine environment[63].

Based on the solid-state $^{31}$P NMR findings of this study, the deposition of Saharan dust enriches the Mediterranean oligotrophic surface waters with P-diester compounds having mainly biological origin (e.g., bacteria communities). The average deposited mass of P-diesters in the Saharan dust aerosols was $0.6 \pm 0.1\ \mu g\ mg^{-1}$ ($n = 4$), contributing 85% to the TP mass. Considering that dust particles were mainly accumulated in the coarse fraction (d > 1.5 μm) and assuming a deposition velocity ($V_d$) of 0.02 m s$^{-1}$, the dry deposition fluxes were calculated as follows: $F = -V_d \times C$, where $F$ is the flux in μg m$^{-2}$ s$^{-1}$, $C$ is the atmospheric mass concentration of dust in μg m$^{-3}$, and $V_d$ is the deposition velocity in m s$^{-1}$. The average daily atmospheric dry deposition of Saharan dust mass ($n = 4$) was $0.4 \pm 0.3$ g m$^{-2}$ d$^{-1}$, which corresponds to the dry daily deposition of $240.5 \pm 208.9$ μg m$^{-2}$ d$^{-1}$ of P-diesters, highlighting the importance of dust as a nutrient source for aquatic environments. Converting the P-diester flux into carbon uptake using the Redfield C/P ratio of 106, we found that the dry daily deposition contributes ~70% to the new productivity of the eastern Mediterranean aquatic environment during a Saharan dust event (new production: 5 g C m$^{-2}$ y$^{-1}$)[64]. The increased duration and severity of Saharan dust episodes in the Mediterranean region as a consequence of global climate change[65,66] is expected to have a significant impact on this P-limited marine environment. A future scenario could be the dust event on March 21, 2018 (Dust-103, Table 1); in this event, the daily average atmospheric particulate mass was 1083 μg m$^{-3}$, five times higher than the typical dust event of the area. This Saharan dust event corresponds to the deposition of 1095 μg m$^{-2}$ d$^{-1}$ of P-diesters and, extrapolating to the eastern Mediterranean surface, is 1524 tonnes d$^{-1}$. Overall, this study demonstrates that the biological material (e.g., bacteria) present in dust is an important vector of organic P, which will eventually be deposited in surface marine waters affecting the biogeochemistry of the Mediterranean environment.

## Data availability

Experimental data are available at https://doi.org/10.5281/zenodo.14876671.

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

## Acknowledgements

This work was supported by the Swiss National Science Foundation, project 8799 (Atmospheric Acidity Interactions with Dust and its Impacts; AAIDI), Swiss National Science Foundation project 215416 (Lipidomics: the new tool in understanding the response of airborne biological particles in a changing atmosphere; LIPIC-AIR), the French National Recherche Agency (FIRETRAC project No ANR-20-CE01-0012-01) and the European Research Council (ERC) project "PyroTRACH" (Grant agreement No. 726165). MK acknowledges support by the Hellenic Foundation for Research and Innovation (HFRI), project BioCAN, grant No 4050. Laura Piveteau is acknowledged for recording a portion of the NMR spectra at the Institute of Chemistry and Chemical Engineering, NMR Platform, École Polytechnique Fédérale de Lausanne, 1015 Lausanne, Switzerland. Dr. Tsagkaraki Maria from the University of Crete is acknowledged for sampling collection in Heraklion. Gordanna Pistoletti (EPFL- ALPOLE, Sion) is acknowledged for her technical contribution to the ONT sequencing of Fungi and Pollen.

## Author contributions

K.V. conceived and designed the study. K.V., C.P., and C.E.A. collected and analyzed data. P.R. and E.A. contributed to the analysis of microbiology data. K.V. C.P., and A.N., synthesized the data and wrote the manuscript. M.K. provided the infrastructure for the sampling station. All authors contributed to editing and discussion of the paper.

## Competing interests

The authors declare no competing interests.
