## [Transparent Peer Review file · Communications Earth & Environment]

Solid-state ^{31}P NMR reveals the biological organophosphorus compounds as the dominant phosphorus species in Saharan dust aerosols

Corresponding Author: Dr Kalliopi Violaki

Version 0:

Decision Letter:

Dear Dr Violaki,

Your manuscript titled "Biological organophosphorus compounds are the dominant P species in Saharan dust aerosols: Evidence by solid-state ^{31}P NMR" has now been seen by 3 reviewers, and we include their comments at the end of this message. They find your work of interest, but some important points are raised. Specifically, more detailed information about the methodology used in order to reproduce the study and an improved discussion to clarify the relevance of the studied samples in the broader context of dust events in this region are needed. We are interested in the possibility of publishing your study in *Communications Earth & Environment*, but would like to consider your responses to these concerns and assess a revised manuscript before we make a final decision on publication.

We therefore invite you to revise and resubmit your manuscript, along with a point-by-point response that takes into account the points raised. Please highlight all changes in the manuscript text file.

Please submit your point-by-point responses as a separate file, distinct from your cover letter where you can add responses to the Editors' comments that you do not want to be made available to the reviewers. Word files are preferred. We recommend that any figures, tables or graphs that are included in the response to reviewers are also included in the main article or Supplementary Information.

Please use the following link to submit your revised manuscript, point-by-point response to the referees' comments (which should be in a separate document to any cover letter), a tracked-changes version of the manuscript (as a PDF file) and the completed checklist:

Link Redacted

We hope to receive your revised paper within six weeks; please let us know if you aren't able to submit it within this time so that we can discuss how best to proceed. If we don't hear from you, and the revision process takes significantly longer, we may close your file. In this event, we will still be happy to reconsider your paper at a later date, as long as nothing similar has been accepted for publication at *Communications Earth & Environment* or published elsewhere in the meantime.

Please do not hesitate to contact us if you have any questions or would like to discuss these revisions further. We look forward to seeing the revised manuscript and thank you for the opportunity to review your work.

Best regards,

Nora Zannoni, PhD

Editorial Board Member
Communications Earth & Environment
orcid.org/0000-0003-2721-5362

Alice Drinkwater, PhD
Associate Editor
Communications Earth & Environment

EDITORIAL POLICIES AND FORMATTING

Editorial Policy: [Policy requirements](https://www.nature.com/documents/nr-editorial-policy-checklist.pdf) (Download the link to your computer as a PDF.)

- Behavioural and social science
- Ecological, evolutionary & environmental sciences
- Life sciences

<https://www.nature.com/documents/nr-reporting-summary.zip>

Furthermore, please align your manuscript with our format requirements, which are summarized on the following checklist: [Communications Earth & Environment formatting checklist](https://www.nature.com/documents/commsj-phys-style-formatting-checklist-article.pdf)

and also in our style and formatting guide [Communications Earth & Environment formatting guide](https://www.nature.com/documents/commsj-phys-style-formatting-guide-accept.pdf).

*** DATA: Communications Earth & Environment endorses the principles of the Enabling FAIR data project (<http://www.copdess.org/enabling-fair-data-project/>). We ask authors to make the data that support their conclusions available in permanent, publically accessible data repositories. (Please contact the editor if you are unable to make your data available).

All Communications Earth & Environment manuscripts must include a section titled "Data Availability" at the end of the Methods section or main text (if no Methods). More information on this policy, is available at <http://www.nature.com/authors/policies/data/data-availability-statements-data-citations.pdf>.

If a community resource is unavailable, data can be submitted to generalist repositories such as [figshare](https://figshare.com/) or [Dryad Digital Repository](http://datadryad.org/). Please provide a unique identifier for the data (for example a DOI or a permanent URL) in the data availability statement, if possible. If the repository does not provide identifiers, we encourage authors to supply the search terms that will return the data. For data that have been obtained from publically available sources, please provide a URL and the specific data product name in the data availability statement. Data with a DOI should be further cited in the methods reference section.

REVIEWER COMMENTS:

Reviewer #1 (Remarks to the Author):

The article claims to a change of paradigm regarding the origin of P related to dust aerosols from North African deserts. While most studies relate P to inorganic sources, the authors demonstrate in this study a dominant organic P signature from microorganisms. The article is well written and the topic is well introduced and contextualized.

The article includes a novel methodology in atmospheric sciences (NMR) to analyse different P forms. The use of the mentioned method has provided interesting results in this study, implying an advancement in the state of the art in this regard.

Conclusions and findings are, in general, well supported and relevant for the scientific community.

Points for discussion

The study and the conclusions are based on a sampling campaign performed in March 2018, which corresponds to the peak season of dust affecting the eastern Mediterranean. Dust particles, according to the trajectories calculated by the authors, are mostly passing over the same region in North Africa, but indicating very long trajectories before and after over marine areas in the Atlantic and Mediterranean, and other continental parts such as the Iberian Peninsula.

Apart of the case with very high concentrations (more than 1200 ug/m³), which concentrations were observed in other cases? This information is important to evaluate the nature/intensity of the samples analysed.

Do the authors consider a contribution (limited, moderate or dominant?) from other sources different to dust? They mention some references to justify their conclusion (41-46), but the match between groups is not straightforward.

In this sense, to what extent their findings/conclusions are extensive to all/most of dust incomes affecting the eastern Mediterranean?

Did the authors compare no-dust situations in their 48 sampling periods against dust events? This should be key to isolate groups of microorganisms only related to desert dust.

Calculations for dust deposition from airborne dust are provided in different studies, but real conditions are usually very different. The comparison between the concentration of dust and the deposition of dust is not straightforward, and in many cases is more controlled by the occurrence of few droplets related to wet deposition rather than a continuous dry deposition process. Therefore, I suggest to disregard or modulate the discussion in lines 223-231. In addition, even in the global change scenario in which we are immersed with intense episodes more frequent over time, the situation claimed in lines 233-234 "A representative future scenario could be the dust event on March 21, 2018 (Dust-103, Table 1)" is just a possibility, but it seems the most possible situation as it is written.

Reviewer #2 (Remarks to the Author):

General Comments

The authors seek fundamental information about the global P cycle. They collected Saharan desert dust samples in the eastern Mediterranean region and analyzed the collections using ³¹P-magic angle spinning NMR to characterize the types (compound classes) of P molecules. They found that P-diesteres accounted for a majority (78-88%) of the total P, followed by orthophosphate (10-19%) and trace concentrations of polyphosphate (1-8%). They attributed the diester dominance to the presence of phospholipids and nucleotides/nucleic acids present in the source (dust) materials. Using e-DNA, they linked the P compounds to specific bacteria including Actinobacteria and Proteobacteria, as shown previously. The results of this study link Saharan dust to eastern Mediterranean Sea productivity through the deposition of bioavailable P. This is a straightforward study that builds on previous observations of dust particles in this region and the potential ecological consequences of atmospheric P deposition.

Specific Comments

1. Most of the e-DNA collected in this study was probably derived from dead/non-viable (relic) microbes. Does this alter the interpretations presented? The authors seem to equate 16S rRNA genes with viable cell numbers.
2. lines 145-146 and lines 203-204: It is important to stress what is novel in the present study.

Reviewer #3 (Remarks to the Author):

This study reports, for the first time, the application of ³¹P solid-state NMR spectroscopy for the characterization and quantification of the organic forms of phosphorus (P) in Saharan dust and deduce whether these African dust emissions could be a significant external P nutrient source for marine environments, namely for the Mediterranean surface waters. Based on the NMR findings, the authors concluded that the deposition of Saharan dust in the Mediterranean oligotrophic surface waters is an important external source of P-diester compounds, with the latter having mainly a biological origin. This study and its main findings are novel and of interest to others in the atmospheric research community and the wider field and, therefore, merits publication. However, there are a few minor additional comments that need to be addressed by the authors before the manuscript can be accepted for publication. These are as follows:

Section 2.1: It is a fact that solid-state NMR (either ³¹P or ¹³C) has the advantage of allowing the analysis of a sample without any previous treatment. The authors describe that the Saharan dust aerosols were collected on pre-combusted quartz filters. We know that the aerosol particles (even Saharan dust aerosols) tend to be trapped within the filter's matrix.

Thus, it would be very helpful if the authors include (or explain) in this section how the dust samples were removed from the filters for subsequent NMR characterization. In line 98, it is stated that samples were packed into 3.2-mm zirconia rotors. Does this mean that the dust samples scraped off from the filters surface to fill in the rotors? If so, doesn't this procedure also remove filter fibres, which are difficult to separate from the collected atmospheric aerosols?

Furthermore, the samples were collected in 2018. Assuming that the ³¹P NMR spectra were acquired recently, it is important to provide additional information on how the dust aerosol samples were stored during this time and the possible degree to which the samples (and thus, the organic forms of P) might have suffered chemical transformations during this long storage period.

Overall, additional information on samples pretreatment and storage for subsequent ³¹P NMR analysis is highly required for the potential reader (experts and non-experts).

Figure 2: It might be helpful for the reader if the authors mention that the ³¹P NMR spectrum of the dust sample is overlaid by the dmfit Monte Carlo simulation. Otherwise, it is difficult to identify the spectrum in the figure (the same is true for the spectra shown in Figure S2).

Lines 199-200: Could the assignment of these plant Orders originate from agricultural activities? It is rather odd the presence of plant Orders associated to broccoli, cauliflower and mustard in Saharan dust aerosols. The authors should provide additional explanations for the supposed presence of these plant Orders in the dust samples.

Communications Earth & Environment is committed to improving transparency in authorship. As part of our efforts in this direction, we are now requesting that all authors identified as 'corresponding author' create and link their Open Researcher and Contributor Identifier (ORCID) with their account on the Manuscript Tracking System prior to acceptance. ORCID helps the scientific community achieve unambiguous attribution of all scholarly contributions. You can create and link your ORCID from the home page of the Manuscript Tracking System by clicking on 'Modify my Springer Nature account' and following the instructions in the link below. Please also inform all co-authors that they can add their ORCIDs to their accounts and that they must do so prior to acceptance.

Version 1:

Decision Letter:

Dear Dr Violaki,

Your revised manuscript titled "Biological organophosphorus compounds are the dominant P species in Saharan dust aerosols: Evidence by solid-state ³¹P NMR" has now been seen by our reviewers, whose comments appear below. In light of their advice we are delighted to say that we are happy, in principle, to publish a suitably revised version in Communications Earth & Environment.

We therefore invite you to revise your paper one last time to address the remaining concerns of our reviewers, which pertain to toning down claims of novelty in your method. At the same time we ask that you edit your manuscript to comply with our format requirements and to maximise the accessibility and therefore the impact of your work.

EDITORIAL REQUESTS:

*****Please take care to match our formatting and policy requirements. We will check revised manuscript and return manuscripts that do not comply. Such requests will lead to delays. *****

Please outline your response to each request in the right hand column. Please upload the completed table with your

manuscript files as a Related Manuscript file.

SUBMISSION INFORMATION:

OPEN ACCESS:

Communications Earth & Environment is a fully open access journal. Articles are made freely accessible on publication. For further information about article processing charges, open access funding, and advice and support from Nature Research, please visit <https://www.nature.com/commsenv/open-access>

Link Redacted

Best regards,

Alice Drinkwater, PhD
Associate Editor
Communications Earth & Environment

REVIEWERS' COMMENTS:

Reviewer #1 (Remarks to the Author):

The authors have considered all my comments provided in the first revision. Some of them were just clarifications to better understand some of the work, the others needed for few inserts/modifications in the text. The few implementations made after my suggestions are suited and make this work clearer.

Reviewer #2 (Remarks to the Author):

The authors have provided adequate responses to the questions that I raised in the initial round of review. In response to my question about 'novelty' the authors claim to be the first to use ^{31}P -NMR in aerosol samples. The method has been used extensively for several decades, as the reviewers acknowledge, for seawater suspended particles, marine sediments and soils. A quick look at the literature uncovered several previous applications to dust/aerosol particulate matter (O'Day et al. 2020, Environmental Science and Technology 54(8):4984) and the extensive work conducted in Chile regarding ^{31}P -NMR analysis of volcanic ash. The authors should be sure to conduct an extensive literature search, or refrain from claiming to 'be the first.' It is more important to stress what they have found and the implications of their work than pioneer status. This is a good paper nevertheless and deserves to be published.

Reviewer #3 (Remarks to the Author):

The revised manuscript has much ameliorated with respect to its first version. This Reviewer has appreciated the way by which the authors have included the suggested amendments in the revised version of the manuscript.

** Visit Nature Research's author and referees' website at <http://www.nature.com/authors> for information about policies, services and author

benefits**

Lausanne, December 19th 2024

Dear all,

Enclosed please find the responses to reviewers' comments for *Communications Earth & Environment* (COMMSENV-24-2776-T): “**Biological organophosphorus compounds are the dominant P species in Saharan dust aerosols: Evidence by solid-state ³¹P NMR**” by Violaki et al. A point-by-point response embedded as blue text are presented below. All our modifications/comments can be easily identified by the editor/reviewers in the revised MS. Thank you again for allowing us to revise our paper and hope that our revision addresses all your concerns.

We look forward to hearing your response.

Best Regards

Kalliopi Violaki

Reviewer #1 (Remarks to the Author):

The article claims to a change of paradigm regarding the origin of P related to dust aerosols from North African deserts. While most studies relate P to inorganic sources, the authors demonstrate in this study a dominant organic P signature from microorganisms. The article is well written and the topic is well introduced and contextualized. The article includes a novel methodology in atmospheric sciences (NMR) to analyse different P forms. The use of the mentioned method has provided interesting results in this study, implying an advancement in the state of the art in this regard. Conclusions and findings are, in general, well supported and relevant for the scientific community.

We greatly appreciate the reviewer's suggestions and their positive comments on the scientific value and importance of our work. We believe that the changes made have fully addressed all his/her concerns.

Points for discussion

1. The study and the conclusions are based on a sampling campaign performed in March 2018, which corresponds to the peak season of dust affecting the eastern Mediterranean. Dust particles, according to the trajectories calculated by the authors, are mostly passing over the same region in

North Africa, but indicating very long trajectories before and after over marine areas in the Atlantic and Mediterranean, and other continental parts such as the Iberian Peninsula. Apart of the case with very high concentrations (more than 1200 $\mu\text{g}/\text{m}^3$), which concentrations were observed in other cases? This information is important to evaluate the nature/intensity of the samples analysed.

During the sampling period, the average concentration of TSP atmospheric particulate matter was estimated gravimetrically. The concentration of atmospheric particulate matter corresponding to each dust event has been added in Fig. S1 in the revised SI.

2. Do the authors consider a contribution (limited, moderate or dominant?) from other sources different to dust? They mention some references to justify they conclusion (41-46), but the match between groups is not straightforward. In this sense, to what extent their findings/conclusions are extensive to all/most of dust incomes affecting the eastern Mediterranean?

P-diester sources are not limited to bioaerosols within dust particles (primarily soil bacteria) but also include other bioparticles identified in the samples, e.g., fungi or plant material and other eucaryotic cells (lines 194-196). In addition to Saharan soil dust, anthropogenic imprints from populated areas of North Africa may also contribute to P-diesters as airborne bacterial communities (Zhao et al., 2022).

3. Did the authors compare no-dust situations in their 48 sampling periods against dust events? This should be key to isolate groups of microorganisms only related to desert dust.

We appreciate the reviewer's excellent suggestion to compare the microbiome of the airborne particles during the dust and non-dust periods; this comparison will be explored in a future study, as it is outside the scope of this paper.

4. Calculations for dust deposition from airborne dust are provided in different studies, but real conditions are usually very different. The comparison between the concentration of dust and the deposition of dust is not straightforward, and in many cases is more controlled by the occurrence of few droplets related to wet deposition rather than a continuous dry deposition process. Therefore, I suggest to disregard or modulate the discussion in lines 223-231.

We agree that wet deposition can dominate over dry deposition, but for the conditions we consider in this study - characterized by high dust concentrations in East Mediterranean during spring - there is no wet deposition acting. This is further supported from meteorological data which indicate that no rainfall occurred during the dust events presented in this study (Table S3). Under this assumption, which represents a lower estimate of daily deposition rate by considering only the dry deposition (in cases of both deposition types were present), we performed the P flux calculations as presented. This point is further clarified in the revised manuscript.

The following sentence (lines 230-231) has been revised in the revised manuscript: "The average *daily* atmospheric *dry* deposition of Saharan dust mass ($n = 4$) was $0.4 \pm 0.3 \text{ g m}^{-2} \text{ d}^{-1}$, which corresponds to the *dry daily* deposition of $240.5 \pm 208.9 \mu\text{g m}^{-2} \text{ d}^{-1}$

of P-diesters, highlighting the importance of dust as a nutrient source for aquatic environments.”

5. In addition, even in the global change scenario in which we are immerse with intense episodes more frequent over time, the situation claimed in lines 233-234 “A representative future scenario could be the dust event on March 21, 2018 (Dust-103, Table 1)” is just a possibility, but it seems the most possible situation as it is written.

To address the reviewer’s concern the word “representative” was removed from the MS.

Reviewer #2 (Remarks to the Author):

General Comments

The authors seek fundamental information about the global P cycle. They collected Saharan desert dust samples in the eastern Mediterranean region and analyzed the collections suing ³¹P-magic angle spinning NMR to characterize the types (compound classes) of P molecules. They found that P-diesters accounted for a majority (78-88%) of the total P, followed by orthophosphate (10-19%) and trace concentrations of polyp (1-8%). They attributed the diester dominance to the presence of phospholipids and nucleotides/nucleic acids present in the source (dust) materials. Using e-DNA, they linked the P compounds to specific bacteria including Actinobacteria and Proteobacteria, as shown previously. The results of this study link Saharan dust to eastern Mediterranean Sea productivity through the deposition of bioavailable P. This is a straightforward study that builds on previous observations of dust particles in this region and the potential ecological consequences of atmospheric P deposition.

We thank the reviewer for the positive feedback on the scientific merit and importance of the work. We hope that all the changes made have now addressed all his/her concerns.

Specific Comments

1. Most of the e-DNA collected in this study was probably derived from dead/non-viable (relic) microbes. Does this alter the interpretations presented? The authors seem to equate 16s rRNA genes with viable cell numbers.

The extracted DNA may indeed originate from highly stressed or dead cells, possibly due to exposure to stressors during atmospheric residence (e.g., UV radiation, atmospheric pollution, low humidity), further exacerbated by the harsh sampling conditions (high-volume sampler). Nonetheless, even if the cells are no longer viable, their DNA remains accessible for quantification and microorganism identification. Therefore, the identification of bioparticles is not compromised.

Furthermore, the 16S rRNA gene is used here not to estimate the number of viable cells but as widely used in the literature as a molecular marker for exploring

evolutionary relationships and profiling microbial composition across diverse environments (Pan et al., 2023).

2. lines 145-146 and lines 203-204: It is important to stress what is novel in the present study.

The novelty of our study is highlighted in lines 28–42, where we summarized that the bioavailable phosphorus (P) in atmospheric dust originates from organic P of biological origin, rather than from inorganic phosphorus found in apatite materials. This breakthrough was achieved through the application of solid-state ^{31}P NMR, a technique used for the first time in atmospheric research. In contrast to previous studies, which inferred organic P by subtracting inorganic P from total phosphorus using colorimetric methods, solid-state ^{31}P NMR directly and simultaneously identifies both inorganic and organic P species (as discussed in lines 72–77 of the introduction). Moreover, our study clearly shows that the organic P in dust is associated with bioaerosols derived from biological material transported with Saharan dust.

Reviewer #3 (Remarks to the Author):

This study reports, for the first time, the application of ^{31}P solid-state NMR spectroscopy for the characterization and quantification of the organic forms of phosphorus (P) in Saharan dust and deduce whether these African dust emissions could be a significant external P nutrient source for marine environments, namely for the Mediterranean surface waters. Based on the NMR findings, the authors concluded that the deposition of Saharan dust in the Mediterranean oligotrophic surface waters is an important external source of P-diester compounds, with the later having mainly a biological origin. This study and its main findings are novel and of interest to others in the atmospheric research community and the wider field and, therefore, merits publication. However, there are a few minor additional comments that need to be addressed by the authors before the manuscript can be accepted for publication. These are as follows:

We thank the reviewer for all the modifications/suggestions and for the positive feedback on the science and importance of the work. We hope now that all the changes made have addressed all his/her concerns.

Section 2.1: It is a fact that solid-state NMR (either ^{31}P or ^{13}C) has the advantage of allowing the analysis of a sample without any previous treatment. The authors describe that the Saharan dust aerosols were collected on pre-combusted quartz filters. We know that the aerosol particles (even Saharan dust aerosols) tend to be trapped within the filter's matrix. Thus, it would be very helpful if the authors include (or explain) in this section how the dust samples were removed from the filters for subsequent NMR characterization. In line 98, it is stated that samples were packed into 3.2-mm zirconia rotors. Does this mean that the dust samples scraped off from the filters surface to fill in the rotors? If so, doesn't this procedure also remove filter fibres, which are difficult to separate from the collected atmospheric aerosols?

We have addressed the reviewer's comment; and the detailed protocol of the sample handling was reported now in the lines 98-100. Specifically, the following sentence has been added:

“Each dust sample was collected by gently scraping the surface of the quartz filter with a clean, sterilized aluminum spoon, avoiding detaching the quartz fibers during the process.”

Furthermore, the samples were collected in 2018. Assuming that the ^{31}P NMR spectra were acquired recently, it is important to provide additional information on how the dust aerosol samples were stored during this time and the possible degree to which the samples (and thus, the organic forms of P) might have suffered chemical transformations during this long storage period. Overall, additional information on samples pretreatment and storage for subsequent ^{31}P NMR analysis is highly required for the potential reader (experts and non-experts).

The dust filters were stored at -20°C after collection, a standard practice for preserving most organic compounds. Chemical analyses for total phosphorus (TP) and for the major ions were conducted within 3 months of sample collection. The solid-state ^{31}P NMR analysis was performed 2 years later; however, the total phosphorus concentrations were comparable between the two techniques, indicating that the storage conditions did not affect the phosphorus species' abundance.

Figure 2: It might be helpful for the reader if the authors mention that the ^{31}P NMR spectrum of the dust sample is overlaid by the dmfit Monte Carlo simulation. Otherwise, it is difficult to identify the spectrum in the figure (the same is true for the spectra shown in Figure S2).

We agree with this comment and we added this information in Fig. S2 legend.

Lines 199-200: Could the assignment of these plant Orders originate from agricultural activities? It is rather odd the presence of plant Orders associated to broccoli, cauliflower and mustard in Saharan dust aerosols. The authors should provide additional explanations for the supposed presence of these plant Orders in the dust samples.

The identification of plant species was revised by using different primers. Still the predominant species in order level are the *Brassicales* ($25 \pm 20\%$) and the *Fabales* ($18 \pm 28\%$). Those broad plants families could be also originated from local agriculture activities, or being long-range transported from North Africa areas.

The lines 200-201 were revised as “Additionally, the majority of identified plant Orders (Fig. S3), belongs to the *Brassicales* ($25 \pm 20\%$) and the *Fabales* Order ($18 \pm 28\%$), which include species from agriculture activities.”

References

Zhao J. et al. Global airborne bacterial community—interactions with Earth's microbiomes and anthropogenic activities Proc. Natl. Acad. Sci. U. S. A., 119, (2022), Article e2204465119.

Pan P, Gu Y, Sun D, Wu QL, Zhou N. 2023. Microbial Diversity Biased Estimation Caused by Intragenomic Heterogeneity and Interspecific Conservation of 16S rRNA Genes. Appl Environ Microbiol 89:e02108-22.

<https://doi.org/10.1128/aem.02108-22>

Lausanne, 14 February 2025

REVIEWERS' COMMENTS:

Reviewer #1 (Remarks to the Author):

The authors have considered all my comments provided in the first revision. Some of them were just clarifications to better understand some of the work, the others needed for few inserts/modifications in the text. The few implementations made after my suggestions are suited and make this work clearer.

We thank the reviewer for the positive feedback on the science and the importance of the work.

Reviewer #2 (Remarks to the Author):

The authors have provided adequate responses to the questions that I raised in the initial round of review. In response to my question about 'novelty' the authors claim to be the first to use ^{31}P -NMR in aerosol samples. The method has been used extensively for several decades, as the reviewers acknowledge, for seawater suspended particles, marine sediments and soils. A quick look at the literature uncovered several previous applications to dust/aerosol particulate matter (O' Day et al. 2020, Environmental Science and Technology 54(8):4984) and the extensive work conducted in Chile regarding ^{31}P -NMR analysis of volcanic ash. The authors should be sure to conduct an extensive literature search, or refrain from claiming to 'be the first.' It is more important to stress what they have found and the implications of their work than pioneer status. This is a good paper nevertheless and deserves to be published.

We would like to express our gratitude to the reviewer for the positive feedback regarding the scientific significance and importance of our work. We fully agree with the reviewer that the primary focus should be on emphasizing the main conclusion of this study, rather than whether this work is the first or second to address the topic. We are aware of the study by O' Day et al. (2020) and we are pleased to note that their findings align with ours, particularly in identifying organic phosphorus (P) as the dominant species in total phosphorus. However, there are notable differences between the studies. O' Day et al. (2020) employed liquid ^{31}P NMR, whereas we utilized solid-state ^{31}P NMR. The liquid ^{31}P NMR method requires sample pretreatment with NaOH, which could potentially hydrolyze diesters into monoesters, complicating the accurate speciation of these two ester forms. In contrast, solid-state ^{31}P NMR does not require such pretreatment. In conclusion, our study remains the only one to use solid-state ^{31}P NMR without pretreatment, and, more importantly, our findings demonstrate that organic phosphorus in dust is associated with biological material transported by Saharan dust.

To address the concerns of the reviewer the following changes were made in the manuscript.
1. The phrase in line 82: "it has never been employed in atmospheric P studies" was replaced by the phrase: "it was not extensively explored in atmospheric phosphorus research".

2. The study O' Day et al. (2020), was cited in line 199 of the manuscript and added in the References section.

Reviewer #3 (Remarks to the Author):

The revised manuscript has much ameliorated with respect to its first version. This Reviewer has appreciated the way by which the authors have included the suggested amendments in the revised version of the manuscript.

We greatly appreciate the reviewer for his/her contribution on the scientific value and importance of our work.